# Stability of Paracetamol Instant Jelly for Reconstitution: Impact of Packaging, Temperature and Humidity

**DOI:** 10.3390/gels8030144

**Published:** 2022-02-25

**Authors:** Samah Hamed Almurisi, Khater AL-Japairai, Farhan Alshammari, Fawaz Alheibshy, Rana M.F. Sammour, Abd Almonem Doolaanea

**Affiliations:** 1Department of Pharmaceutical Technology, Kulliyyah of Pharmacy, International Islamic University Malaysia (IIUM), Kuantan 25200, Malaysia; samahhamed8611@gmail.com; 2Department of Pharmaceutical Engineering, Faculty of Chemical and Process Engineering Technology, Universiti Malaysia Pahang, Gambang 26300, Malaysia; khater.11@hotmail.com; 3Department of Pharmaceutics, College of Pharmacy, University of Hail, Hail 2240, Saudi Arabia; frh.alshammari@uoh.edu.sa (F.A.); fa.alheibshy@uoh.edu.sa (F.A.); 4Department of Pharmaceutics, College of Pharmacy, Aden University, Aden 6075, Yemen; 5Pharmaceutics Department, Dubai Pharmacy College for Girls, Dubai 19099, United Arab Emirates; rana@dpc.edu; 6IKOP Sdn Bhd, Kulliyyah of Pharmacy, International Islamic University Malaysia, Kuantan 25200, Malaysia

**Keywords:** paracetamol, stability, packaging, environmental conditions, shelf life

## Abstract

The stability of the medicinal product is a major concern in the pharmaceutical industry and health authorities, whose goal is to guarantee that drugs are delivered to patients without loss of therapeutic properties. This study aims to evaluate the effect of environmental conditions and packaging on the stability of paracetamol instant jelly sachets based on both chemical and physical stability. The paracetamol instant jelly was packaged in plastic sachets (packaging 1) and sealed aluminium bags in screw-capped amber glass bottles (packaging 2), which were stored in real-time and accelerated stability chambers for 3 months. Samples were taken out from the chambers and were characterised for appearance, moisture content, texture, viscosity, in vitro drug release, paracetamol content, and 4-aminophenol level at different time points. The real-time storage condition at a lower temperature maintained the stability of the paracetamol instant jelly, while the accelerated condition led to a significant change in the formulation properties. In addition, the proper packaging of paracetamol instant jelly maintained the paracetamol’s stability, regardless of environmental conditions, for three months. The results show that the environmental conditions and packaging play a significant role in maintaining paracetamol instant jelly stability.

## 1. Introduction

The bitter taste of paracetamol causes a burning sensation in the throat or mouth, which affects paediatric patient compliance. The liquid dosage forms are the most predominant oral paracetamol products commercially available for paediatrics, as they are easy to swallow in comparison to solid products. Nevertheless, liquid dosage forms have greater stability concerns due to the presence of water, which increases the potential for drug degradation, excipient degradation, microbial contamination, and drug–excipient and drug–container interactions. A further significant burden correlated with liquids is the unpleasant taste issues and the requirement for taste masking [1]. For liquid dosage forms of commercially available dosage forms, taste masking is achieved by adding high concentrations of sugar, sweeteners, and flavours with limited success.

Multiparticulate dosage forms such as beads are described as mainly solid oral dosage forms comprised of discrete spherical units. They are considered a flexible solid dosage form for the delivery of drugs to a broad range of patients, including paediatrics, geriatrics, and patients with swallowing difficulties and can be utilised to modify drug release and enhance bioavailability via reducing size, as well as to mask the unpleasant taste of the drug. In addition, beads offer the classic advantages of solid dosage forms over liquid preparations, such as improved stability and reduced weight and bulkiness, resulting in easier transportation, as well as accurate dosing [2]. In previous work, paracetamol was encapsulated in beads and sprinkled in dried jelly for reconstitution to overcome the grittiness and rough mouthfeel perception associated with beads and improve the palatability and patient acceptability [3].

The manufacturing of beads in this study was performed using an electrospray lab set-up using one nozzle. This is a critical part of manufacturing that is not yet well-established in the pharmaceutical industry. Therefore, the electrospray set-up was then scaled up to a pilot manufacturing scale, using 144 nozzles in a GMP manufacturing environment (IKOP SDN BHD, Kuantan, Malaysia), with a grant supported by the Malaysian Ministry of Science, Technology and Innovation (MOSTI, grant number SR1017Q1038, Malaysian patent application number UI2021006238). The possibility of manufacturing was considered starting from the lab-scale batch, as mentioned in our previous study where we evaluated the feasibility of the approach to increase the encapsulation efficiency [3]. After the beads are manufactured, they will be mixed with other excipients to form the dry powder using established commercially available mixers. It is expected that the cost of production will be comparable to the current manufacturing cost of liquid dosage forms.

Stability study of a drug product is very important to ensure its quality over the shelf life under the impact of various environmental factors such as temperature and humidity. The major aim of the pharmaceutical stability testing is to supply sensible confirmation that the products will stay at a satisfactory quality level when reaching the marketplace and until the patient uses the last unit of the product [4]. In general, the stability study is divided into an accelerated stability test (40 ± 2 °C/75 ± 5% RH) for 6 months, intermediate study (30 ± 2 °C/65 ± 5% RH) for 6 months, and long-term study (25 ± 2 °C/60 ± 5% RH or 30 ± 2 °C/65 ± 5% RH) for 12 months or longer [5]. The exact storage conditions, time points for testing, and study duration differ based on the climate zones as defined by the ICH guidelines and might be subject to local authority regulations.

The protocol for stability testing may be different for the same product if it is planned to be distributed or marketed in diverse regions of the world. The Association of Southeast Asian Nations (ASEAN, Jakarta, Indonesia) includes Brunei Darussalam, Cambodia, Indonesia, Lao PDR (Laos), Malaysia, Myanmar, Philippines, Singapore, Thailand, and Vietnam, which belong to the hot and humid climatic zone (zone IV). In order to create a shared market for their pharmaceutical products, ASEAN regulatory authorities have specified harmonised stability testing conditions for marketing authorisation for pharmaceuticals. After consultation and several meetings between regulators and experts from ASEAN countries, the stability conditions stated in the WHO and ICH guidelines cited above did not effectively address the climatic conditions prevalent in the majority of ASEAN countries. Therefore, the conditions were then adopted for stability studies in ASEAN countries according to a meeting held in Jakarta on 12-13 January 2004 [6], as presented in Table 1.

Paracetamol is significantly affected by the presence of moisture: an increase in the temperature usually increases the degradation rate of the drug [7,8,9]. In addition, the stability of paracetamol has been investigated earlier in different brands of paracetamol tablets to investigate if the paracetamol is photodegradable or not. According to the findings of this study, paracetamol is almost photodegradable but undergoes little change in its physio-chemical properties when exposed to direct sunlight [10]. However, the storage of drugs under LED lighting has become more common in recent years, and some drugs, including paracetamol, display a different degree of colour change depending on the light source. Therefore, a brown light-shielding plastic bag provides more protection than a normal plastic bag for the prevention of the colour change of medicines stored under LED lighting [11]. Moreover, pharmaceutical products are obtained as mixtures of active molecules and several excipients, which control the workability, stability, and release properties of the formulations. When paracetamol was mixed with other ingredients, the stability changed based on the possible reactions between paracetamol and excipients, as well as between excipients themselves, especially when we have reactive functional groups such as amine groups (chitosan) and hydroxyl groups (paracetamol, alginate). Therefore, the accurate estimation of a drug’s shelf life is challenging [12].

The packaging is responsible for providing life-saving drugs, from the time of production until their use [13]. The packaging material must be selected based on the specific needs for the protection of the individual products. If the active ingredient is sensitive to humidity or oxygen, contact with the external environment can cause instability and loss of efficacy of the active ingredients. This problem might be solved by storing the drugs away from environmental conditions such as humidity and oxygen or using material that is impermeable to water vapour and oxygen [14,15]. Glass, plastic, paper and paperboard, metal, and aluminium foil are the most commonly used pharmaceutical packaging materials for solid dosage forms. The primary packaging for the pharmaceutical product should sufficiently protect the product from the environment as well as be compatible with the product, while the secondary package, which usually contains multiple primary packages, provides the strength for stacking in the warehouse [16].

The instability of a pharmaceutical product may affect the therapeutic effect by lowering the actual administered dose to the patient and may produce toxic degradation products [4,17]. The assay of a drug substance is necessary to determine the drug content in the dosage form [18]. Thus, this stability study was performed to investigate the effects of different storage temperatures, relative humidity (RH), and packaging on the characteristics of paracetamol instant jelly (single dose sachet). The shelf life of a drug product was determined based on the time a product remains within specifications agreed upon with the regulatory agencies. In general, the specifications of the stability aspect were divided into chemical and physical stability. Chemical stability involves how long a drug product continues to have adequate potency in its packaging, while physical instability was associated with any changes to the drug product performance or appearance [19].

The paracetamol instant jelly in this study was prepared in dry form for reconstitution (single dose). The main reasons for this type of dosage form are to avoid chemical and physical stability problems, as well as to decrease the final product weight, because the aqueous vehicle is absent and, accordingly, transportation expenditures may be lessened [20].

For the stability study, the samples were filled in two forms of packaging: the first was plastic sachets (packaging 1), and the second was sealed aluminium bags in screw-capped amber glass bottles (packaging 2). Both were stored in a real-time (30 °C and 75% RH) and accelerated (40 °C and 75% RH) stability chamber. At different stability time points (0, 2 weeks, 1, 2, and 3 months), samples were taken out from the chambers and were characterised for appearance, moisture content, texture, viscosity, in vitro drug release, paracetamol content, and 4-aminophenol level.

## 2. Results and Discussion

### 2.1. HPLC Method Validation for Quantification of Paracetamol

The system suitability test was conducted by analysing the standard solution of paracetamol (10 µg/mL) in six replicates. The retention time was about 3.37 ± 0.004 min. The developed HPLC method was found to be specific, as it was able to differentiate and quantify paracetamol in the presence of other excipients. There is no peak for blank and placebo at the characterisation peak for paracetamol as observed in Figure 1.

Forced degradation is a process that involves degradation of drug products and drug substances at conditions more severe than accelerated conditions for stability; it thus generates degradation products that can be studied to determine the stability of the molecule [21]. It is preferable to perform forced degradation studies earlier in the drug development process to gain more information about the stability of the molecule in order to enhance the formulation manufacturing process and to determine the storage conditions [22]. The forced degradation is performed under different conditions: light, thermal, acidic, basic, and oxidation. For comparison, the control standard and sample solutions were kept at 4 °C for 1 day to avoid any degradation that might occur. Paracetamol proved to have stability under light stress condition, and no extra peak was observed. On the other hand, the degradation occurred extensively under thermal stress without degradation products, while under acid, base, and oxidative conditions, degradation products were observed, as illustrated in Figure 2.

The method was found to be linear within the concentration range of 1–20 µg/mL for paracetamol, with a regression equation of y = 26,853x + 1362.6 that showed an excellent correlation coefficient of 0.9992, where y and x represent the area under the curve and the concentration of paracetamol in µg/mL, respectively, as shown in Figure 3. The LOD and LOQ of paracetamol were 0.049 µg/mL and 0.14 µg/mL, respectively.

The precision of the analytical procedure was evaluated by determining the intra- and inter-day precision of 100% spiked solution. An analytical method is regarded as more accurate when its recovery is between 98% and 102%, and the intra- and inter-day precision assays were expressed as relative standard deviation (RSD) 1.4 and 1.19, respectively, indicating that the method presents a good precision (RSD ≤ 2%), as presented in Table 2. This study was found to be satisfactory. Therefore, this analytical method could be used for the routine analysis of paracetamol in a pharmaceutical formulation.

### 2.2. Physical Observation and Moisture Content

As illustrated in Figure 4, the appearance of the paracetamol instant jelly in dry form was unchanged for the samples stored under real-time conditions, regardless of packaging used. On the other hand, the colour of the samples changed to brownish-yellow at 1 month and a much darker brownish-yellow at 2 and 3 months in packaging 1 in the accelerated stability chamber. In addition, the moisture content was significantly higher in samples stored in packaging 1 compared with packaging 2 (*p*-value < 0.05), as illustrated in Table 3. The colour change of samples stored in packaging 1 in the accelerated condition indicates the deterioration of their commercial value and quality [23], and it might be attributed to storage conditions such as temperature and moisture [24], as well as the low protection of packaging 1. By combining the results of appearance and moisture content, the change of colour occurred in packaging 1 when the moisture content exceeded 10%. On the other hand, packaging 2 provided more protection against the passage of moisture. The selection of packaging can be one of the most important decisions in the development of a drug because packaging plays a key role in protecting a drug product from the RH of the environment [19]. 

### 2.3. Texture Profile Analysis (TPA)

Hardness (also known as firmness) is the most important sensory characteristic in TPA analysis. It is defined as the force required to achieve a particular deformation and is used to evaluate the mouthfeel [25]. Cohesiveness refers to the strength of the internal structure, and the jelly is easy to chew and swallow when it has a low cohesiveness value [26]. Adhesiveness or stickiness refers to the amount of effort necessary to overcome the attraction forces that exist between the food’s surface and the surface of the material with which it comes into contact, such as tongue, teeth, and palate [27]. Moreover, gumminess is a secondary parameter that describes the energy needed to disintegrate a semi-solid food product. The literature shows that the gumminess of a product increases when hardness increases [27]. The TPA analysis of paracetamol instant jelly was studied previously and compared with commercial products for paediatric use [3]; the results showed that the TPA of paracetamol instant jelly was acceptable and similar to the commercial products used by the paediatric population. For the stability study, the TPA of paracetamol instant jelly samples was evaluated, as illustrated in Figure 5. The TPA of paracetamol instant jelly was stable under the real-time condition for both types of packaging, as well as under the accelerated condition for samples stored in packaging 2. On the other hand, the hardness, adhesiveness, and gumminess were significantly lower (*p*-value < 0.05) compared to the 0 time point sample for the samples stored in packaging 1 under the accelerated conditions after 2 and 3 months. This might be due to higher moisture content; water molecules act as a plasticiser and make products less elastic and more vulnerable to fracture upon compression [28,29].

### 2.4. Flow Behaviour Measurement

The flow behaviour of the paracetamol instant jelly followed shear-thinning behaviour and fit the Ostwald de Waele model with high correlation coefficients of 0.99 and a consistency index K (the indicator of the viscous nature of gelling agents) of 36,370 mPa.s at 0 time point. Most of the flow behaviours for the different types of packaging in the real-time and accelerated stability study at different time points were similar to the 0 time point, as presented in Figure 6, and fit the Ostwald de Waele model with high correlation coefficients of 0.99, as well with consistency index K ranging between 36,370 to 39,740 mPa.s. Nevertheless, this behaviour was not displayed in the samples stored in packaging 1 in the accelerated chamber at 2 and 3 months, for which the correlation coefficients reduced to 0.96 in 2 months and 0.85 in 3 months. The consistency index K also decreased to 28,840 mPa.s in 2 months and to 25,310 mPa.s in 3 months. Overall, the viscosity of the jelly reduced significantly in 2 and 3 months for samples stored in packaging 1 in the accelerated chamber compared to the 0 time point, as a result of the beads sediment in the bottom of the glass, while the beads were distributed in jelly with an absence of sedimentation at the 0 time point (Figure 7). Insufficient viscosity of the jellies leads to the sedimentation of particles, which results in inaccuracy of drug dose, poor acceptance by the patient, and unpleasant scraping sensation in the mouth or throat when swallowing [30,31,32].

### 2.5. In Vitro Drug Release

Dissolution stability is a critical parameter in determining the performance and defining the quality of solid oral products. The drug release for samples stored at packaging 1 in real-time stability chambers was unchanged, while for samples stored in the accelerated stability chambers, the drug release was higher in 2 and 3 months, with no significant difference except in 15 min (*p*-value < 0.05), which was attributed to a greater diffusion of paracetamol from the lower viscosity of these jellies. On the other hand, the drug release from samples stored in packaging 2 at different time intervals was similar to the 0 time point in the real-time and accelerated stability chambers, as presented in Figure 8.

### 2.6. Paracetamol Content

The paracetamol contents (% *w*/*v*) at different stability study points and in different packaging are presented in Table 4. All results pass the USP specification, which is 90 to 110% (*w*/*v*), except for samples stored in packaging 1 in the accelerated condition.

As illustrated in Table 5, the shelf life of the samples stored in packaging 2 was higher than in packaging 1, and the same was seen in real-time storage when the temperature was higher than the accelerated storage temperature. The shelf life must be assigned in the development and commercialisation of a pharmaceutical product to determine how long a product will be safe and effective for the patient under reasonable storage conditions by performing the appropriate assay in real time until the threshold is reached [33]. In general, several pharmaceutical products have a shelf life of 2 years or more, and a 2-year shelf life is usually desirable for any product intended for a global market [34].

### 2.7. 4-Aminophenol Level

No peak of 4-aminophenol appeared in the chromatograms for samples stored in packaging 1 in the real-time chamber as well as the samples stored in packaging 2 at both real time and accelerated stability chambers. On the other hand, the sample that was stored in packaging 1 in the accelerated chamber shows a 4-aminophenol level exceeding the 50 ppm limit, as illustrated in Figure 9. The 4-aminophenol should be kept low in the paracetamol-containing drug product because it can cause hepatotoxicity and nephrotoxicity [35]. In a study conducted in Nigeria on paracetamol tablets that were obtained from the open markets, street vendors, and patent medicine stores, a higher content of para-aminophenol was found in samples. The storage and handling conditions in several retail outlets can be poor and far from the ideal recommendations of USP and BP, which state that paracetamol must be preserved in tight, light-resistant containers stored at room temperature and protected from moisture and heat [36]. Moreover, it has been reported that the shelf life of commercial paracetamol tablets decreases to 11.3% and 31.9% due to hydrolytic degradation to para-aminophenol following an increase in storage temperature from 25 to 37 °C at relative humidity of 75% and 100%, respectively [37].

## 3. Conclusions

The packaging material and environmental conditions must be considered when preparing a paracetamol dosage form as instant jelly. Packaging plays several roles in improving or worsening the stability and shelf life of products because it alters the movement of volatile/gaseous materials between the inside and outside of the packaging. Moreover, the higher humidity and temperature will accelerate the physiochemical changes of paracetamol instant jelly. This study evaluated the impact of packaging and environmental conditions on the stability of paracetamol instant jelly for reconstitution. Two types of packaging were used to represent a low-protection container, such as plastic sachets, and a high-protection one, such as packaging 2, under ambient and accelerated temperature, as well as elevated humidity, which is commonly present in Asian countries. The results indicated that packaging 2 was better than packaging 1, especially under accelerated conditions. Paracetamol instant jelly for reconstitution is very sensitive to moisture and must be protected in packaging that will protect it from harsh environmental conditions or stored in places with low temperature and humidity to maintain the stability of the formulation and provide a longer shelf life.

## 4. Materials and Methods

### 4.1. Materials

Micronised paracetamol was obtained from ZhengjianKangle Pharmaceutical Co., Ltd. (Wenzhou, China). Sodium hydroxide was obtained from Suvidhinath Laboratories. Hydrogen peroxide 35% *w*/*v* and formic acid were supplied by Merck Millipore (Burlington, MA, USA), and methanol (HPLC grade) was purchased from Fisher Scientific Inc. (Selangor, Malaysia). The KBF 240 accelerated and Max 1400 real-time stability chambers were purchased from Capromax (Selangor, Malaysia).

### 4.2. Methods

#### 4.2.1. Stability Study Protocol

The paracetamol instant jelly for this stability study was prepared in a study conducted by Almurisi SH et al. [3], in which the instant jelly formulation consisted of chitosan-coated paracetamol alginate beads, ι-carrageenan (gelling agent), glycine (buffering and sweetening agent), and calcium lactate gluconate (gelation inducer for ι-carrageenan). For the reconstitution method, the dry paracetamol instant jelly was reconstituted with 20 mL water and agitated by hand, stirring for 20 s. The paracetamol instant jelly (single dose) was packed in two types of packaging: the first was sealed plastic sachets (packaging 1), and the second was sealed aluminium bags in screw-capped amber glass bottles (packaging 2), as illustrated in Figure 10. The first packaging option of plastic sachets (primary packaging) was used to observe the ability of plastic to protect the formulation from the environment, since the plastic sachets are considered as cost-effective packaging and are easily sourced from the local market. The second option was sealed aluminium bags (as primary packaging) in screw-capped amber glass bottles (secondary packaging). The purpose of this selection was to provide the most protection for the formulation against light, humidity, and temperature and to observe the role of packaging in maintaining the stability of the formulation under different environmental conditions. In fact, the second packaging option was selected after we observed indications of failure of the first option to provide sufficient protection. The samples were stored in a real-time (30 °C and 75% RH) and an accelerated (40 °C and 75% RH) stability chamber. At the designated stability time points (0, 2 weeks, 1, 2, and 3 months), the samples were taken out and tested.

#### 4.2.2. Analytical Method Validation of Paracetamol

Method validation was performed using high-performance liquid chromatography (HPLC) method. The HPLC system (HPLC Prominence LC20A, Shimadzu Corporation, (Kyoto, Japan) consisted of an LC-20 AD pump, a DGU-20A5R on-line degasser, a SIL-20 an auto injector, and a CTO-20 AD column oven, and an SPD-M20A PDA detector was used for this analysis. The column used was ZORBAX Eclipse Plus C18 4.6 × 250 mm column with pore size of 5 µm (Agilent Technologies). The flow rate was 1.5 mL/min, and the column temperature was set at 25 ± 2 °C. The detection wavelength was 243 nm, with a running time of 12 min per sample. The mobile phase consisted of a mixture of water and methanol (3:1, *v*/*v*). The paracetamol standard solution and sample stock solution were prepared with a concentration of 10 µg/mL.

The standard solution (10 µg/mL) was injected six times to determine the suitability and effectiveness of the chromatographic system prior to use. In the specificity test, blank, standard solution, sample solution, and placebo solution (placebo of jelly was prepared by mixing all the components that contain blank chitosan-coated alginate beads in jelly without adding paracetamol) were compared in term of characterised peak to ensure that no peak appeared for the blank and the placebo samples at the characteristic peak of paracetamol. In addition, if any other peak appeared in the chromatogram, then all the peaks need to be separated, so that there will be no interference on the peak of paracetamol. In order to evaluate the suitability of the method as a stability-indicating HPLC method, stress studies were carried out under ICH recommended conditions. The ICH guideline states that stress testing is intended to identify the likely degradation products, which further helps in the determination of the intrinsic stability of the molecule and establishing degradation pathways and to validate the stability-indicating procedures used (ICH Guideline, 2003). Intentional degradation was attempted by exposing the standard solution (10 µg/mL) and sample solution (10 µg/mL) to the following stress conditions: acid (3 M HCL at room temperature for 2 h), base (1 M NaOH at room temperature for 2 h), peroxide degradation (35% H_2_O_2_ at room temperature for 2 h), thermal degradation (90 °C), and ultraviolet light degradation (24 h). The ability of the proposed methods to measure the analyte response in the presence of its degradation products was studied (Alam, Khanam, Ganguly, Barik, & Siddiqui, 2014; Kamble & Singh, 2012).

The linearity study was carried out with seven concentrations levels, which were 1, 2, 3, 4, 10, 15, and 20 µg/mL, the concentration of paracetamol must fall within the linearity change. A peak area vs. concentration was plotted, on which the equation and the R^2^ were established. Furthermore, the LOD and LOQ were measured by injecting the lowest detectable concentration in the linearity curve 6 times, and the standard deviation was calculated. Based on that standard deviation and curve slope, the LOD and LOQ of the analyte were mathematically estimated using the following equations [38]. To check precision, one concentration level (10 µg/mL) of paracetamol was spiked in the sample solution and injected six times; the concentration level is hereby termed as spiking solution. The precision of spiked solution was determined by repeatability (intra-day) and intermediate precision (inter-day). The intra-day precision was calculated as the relative standard deviation (RSD) of results from six samples during the same day, and the inter-day precision on two different days. The precision is expressed as the percent relative standard deviation (% RSD), which shows accuracy as a percent of the recovery.

#### 4.2.3. Physical Observation

The appearance of the paracetamol instant jelly was determined organoleptically in comparison to the original samples.

#### 4.2.4. Moisture Content

The moisture content of the paracetamol instant jelly powder was measured using an oven-drying method. Three samples (each approximately 1.22 g) were transferred into an oven and dried at 105 °C until a constant weight was achieved. The residual moisture content (loss on drying) was determined as the ratio of the moisture loss weight to the sample weight, and the result was expressed as a percentage [39].

#### 4.2.5. Texture Profile Analysis (TPA)

Texture profile analysis (TPA) of paracetamol instant jelly was evaluated after it was reconstituted in of water in term of hardness, cohesiveness, adhesiveness, and gumminess using CT3 Texture Analyser (Brookfield Laboratories, Middleboro, MA, USA). TPA was set with a trigger of 10 g, deformation of 5 mm, and a speed of 2.0 mm/s. The test consists of two cycles of compression. In each cycle, the sample in each container was penetrated 5 mm by a cylinder probe (3.5 mm diameter) at a crosshead speed of 2 mm/s, and the probe was withdrawn from the sample at the same speed.

#### 4.2.6. Flow Curve Measurement

The viscosity and flow curve of paracetamol instant jelly was evaluated after reconstitution in water using a HAAKE Mars rheometer from Thermo Scientific (MA, USA). The conditions include a measuring temperature of 25 °C and a cone rotor (cone angle: 1°, cone diameter: 35 mm) with a gap of 1 mm. The flow behaviour was measured using the viscosity curve, and the graphs were then presented as viscosity (mPa.s) as a function of the shear rate ($\dot{\gamma }$). Experimental flow curves were fitted to a power law model as follows [40]:(1)σ=κγn where σ is the shear stress (mPa), γ is the shear rate (s^−1^), κ is the consistency index (mPasn), and n is called the flow index. If n is less than 1, the material is shear-thinning, and if n is more than 1, the material is shear-thickening (19). The r value was used to evaluate the goodness of fit to the experimental results in the Ostwald de Waele model.

#### 4.2.7. In Vitro Drug Release

The dissolution of paracetamol instant jelly was carried out in 900 mL of a buffer solution with pH 5.8 at a paddle speed of 50 rpm. At scheduled intervals (15, 20, 30, 45, 60, and 120 min), 3 mL samples were taken, and an equivalent volume of the fresh dissolution medium was added. The aliquots, following suitable dilution, were analysed for drug content at 243 nm using a spectrophotometer Hitachi spectrophotometer model U-1900.

#### 4.2.8. Determination of Drug Content and Shelf Life

The sample solution was prepared by dissolving paracetamol instant jelly containing 200 mg paracetamol in 200 mL of mobile phase to get a nominal concentration of 10 µg/mL of paracetamol. The HPLC system (HPLC Prominence LC20A, Shimadzu Corporation, Japan) was used for this analysis. The order of degradation was determined by the graphical method: the log of % drug remaining for first order reaction was plotted against time at each temperature for samples stored in packaging 1 and packaging 2. The reaction rate constant *K* was measured for each temperature from the slope of log drug remaining vs. time graphs, using the formula given below [41]:(2)$$Slope=\frac{-K\;deg}{2.303}$$

In addition, the shelf life (the time necessary for the drug to decay to 90% of its original concentration) was calculated by substituting *K* deg for each temperature in the following equation [42,43]:(3)$$t_{90\%}\;=\frac{0.1052}{K}$$

#### 4.2.9. 4-Aminophenol Level

A sample solution was prepared by accurately transferring paracetamol instant jelly equivalent to 400 mg of paracetamol into a 50 mL volumetric flask (concentration 8 mg/mL). Additionally, the sample solutions were prepared spiked with 25 µg/mL of 4-AP, that is, corresponding to the 50 ppm limit. For 4-aminophenol determination, the mobile phase consisted of sodium 1-butanesulfonate in a mixture of water and methanol and formic acid (15:85:0.4), and the flow rate was 2.0 mL/min. The detection wavelength was set to 272 nm, with a running time of 10 min per sample [44].

#### 4.2.10. Statistical Analysis

All results were taken in triplicate. Statistical analysis of data was performed using Minitab 17 software (Minitab Inc., State College, PA, USA) by one-way analysis of variance (ANOVA) and a post-hoc Tukey’s test, assuming a confidence level of 95% (*p*-value ≤ 0.05) for statistical significance. All data were presented as mean ± standard deviation.

## Figures and Tables

**Figure 1 gels-08-00144-f001:**
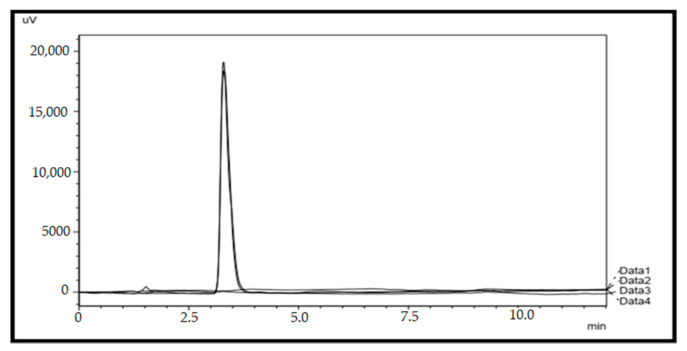
Chromatograms of blank (Data 1), placebo (Data 2), standard (Data 3), and sample (Data 4).

**Figure 2 gels-08-00144-f002:**
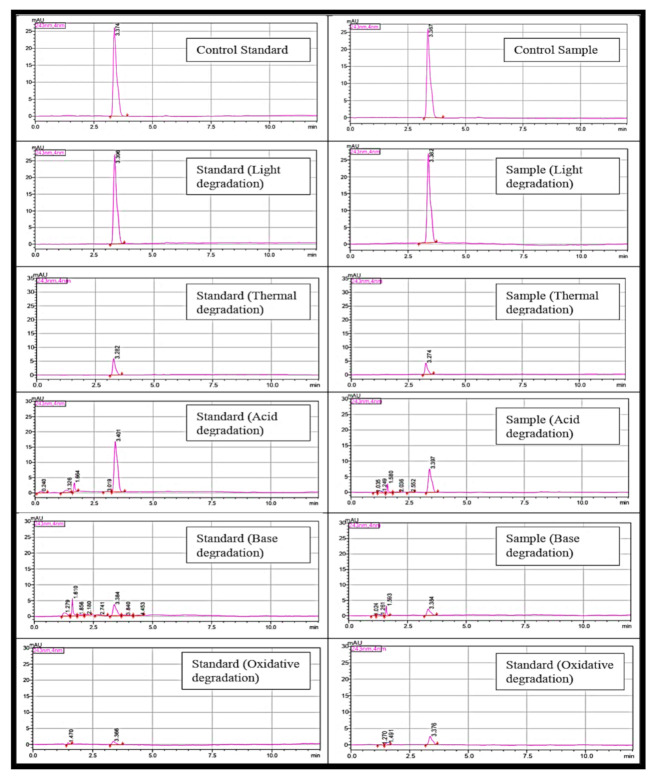
Chromatograms of standard and sample at control and after forced degradation.

**Figure 3 gels-08-00144-f003:**
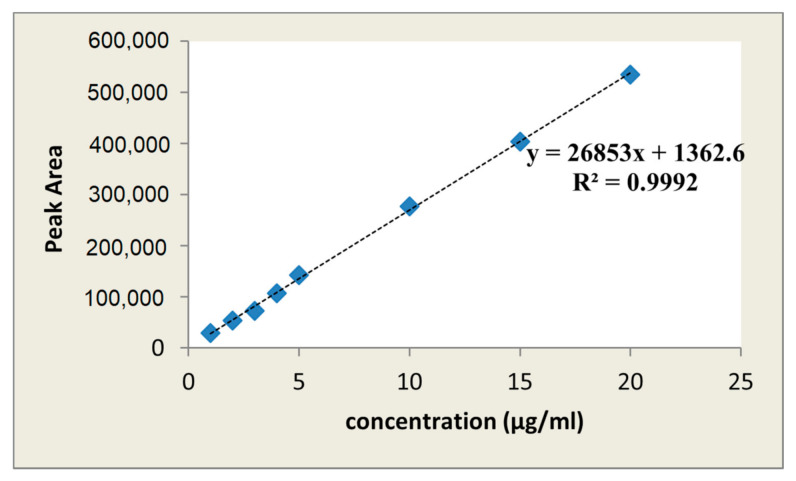
Standard curve of linearity for paracetamol.

**Figure 4 gels-08-00144-f004:**
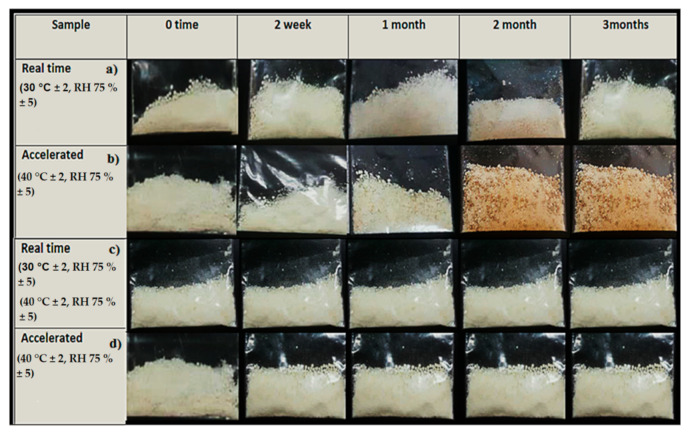
The physical appearance of paracetamol instant jelly samples stored in packaging 1 under (**a**) real-time conditions and (**b**) accelerated conditions and samples stored in packaging 2 under (**c**) real-time conditions and (**d**) accelerated conditions at different time points (0, 2 weeks, 1, 2, and 3 months).

**Figure 5 gels-08-00144-f005:**
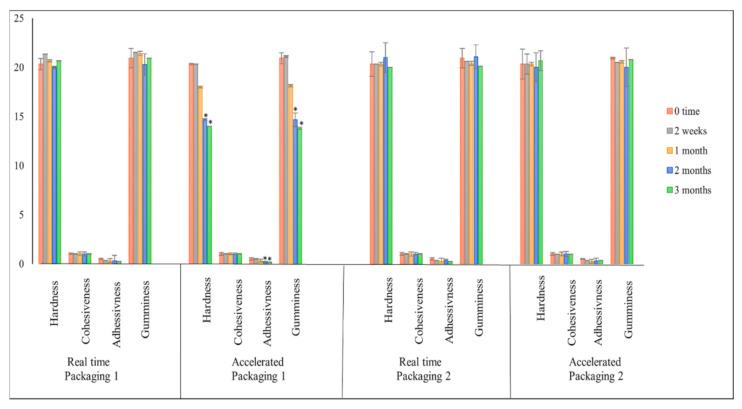
The texture profile analysis (TPA) of paracetamol instant jelly samples stored in packaging 1 and packaging 2 at different time points (0, 2 weeks, 1, 2, and 3 months). Error bars represent standard deviation, * symbol indicate significant differences (*p*-value < 0.05).

**Figure 6 gels-08-00144-f006:**
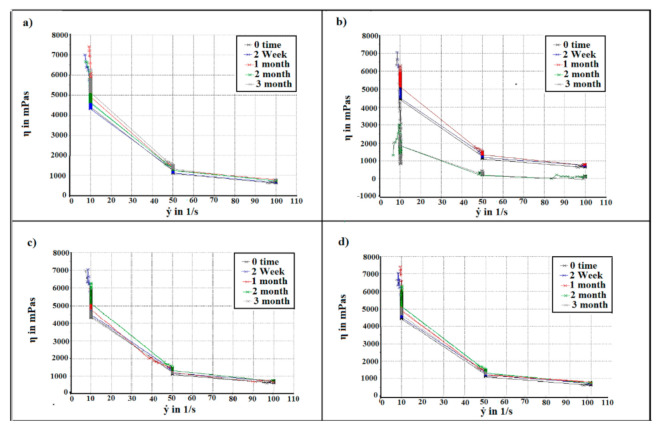
The flow curve of paracetamol instant jelly samples stored in packaging 1 under (**a**) real-time conditions and (**b**) accelerated conditions and samples stored in packaging 2 under (**c**) real-time conditions and (**d**) accelerated conditions at different time points (0, 2 weeks, 1, 2, and 3 months).

**Figure 7 gels-08-00144-f007:**
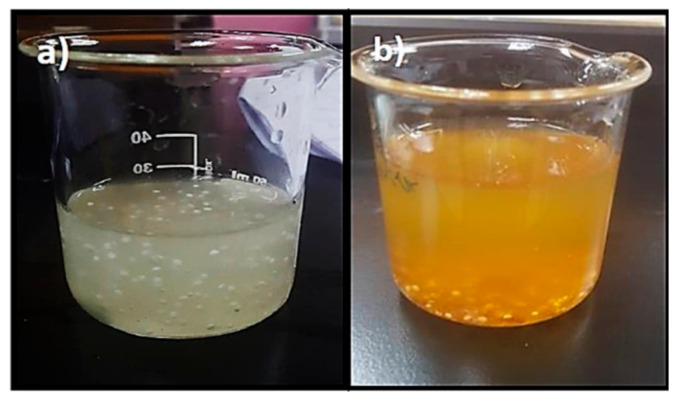
The appearance of jelly: (**a**) at 0 time point and (**b**) after being stored in packaging 1 in the accelerated chamber for 3 months.

**Figure 8 gels-08-00144-f008:**
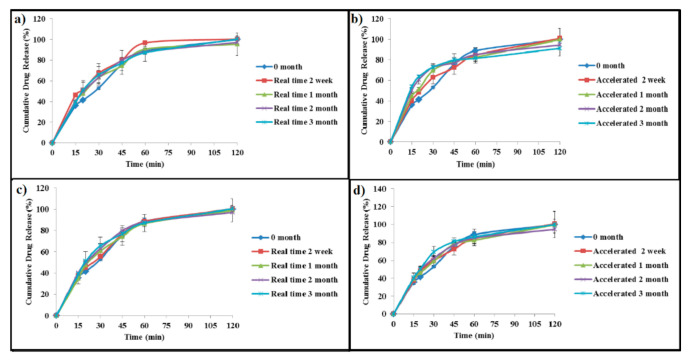
In vitro drug release profile of paracetamol instant jelly samples stored in packaging 1 under (**a**) real-time conditions and (**b**) accelerated conditions and samples stored in packaging 2 under (**c**) real-time conditions and (**d**) accelerated conditions at different time points (0, 2 weeks, 1, 2, and 3 months).

**Figure 9 gels-08-00144-f009:**
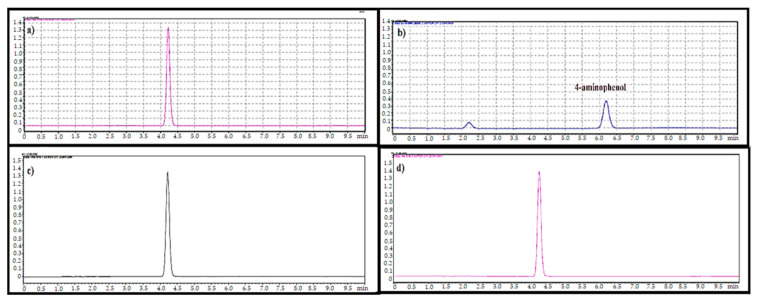
The 4-aminophenol peak in paracetamol instant jelly samples stored in packaging 1: (**a**) no peak in real-time conditions at 0 time point, (**b**) 4-aminophenol peak present in accelerated conditions at 3 months. Samples stored in packaging 2: (**c**) no peak in real-time conditions at 0 time point, (**d**) 4-aminophenol peak present in accelerated conditions at 3 months.

**Figure 10 gels-08-00144-f010:**
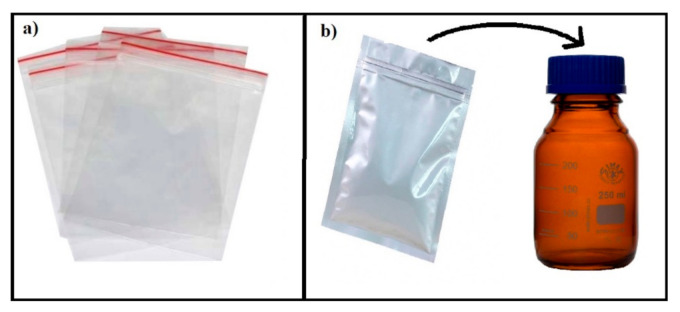
Packaging of paracetamol instant jelly: (**a**) packaging 1 and (**b**) packaging 2.

**Table 1 gels-08-00144-t001:** Stability testing conditions in ASEAN countries.

Type	Conditions
Products in primary containers permeable to water vapour	30 ± 2 °C/75 ± 5% RH
Products in primary containers impermeable to water vapour	30 ± 2 °C/RH not specified
Accelerated studies	40 ± 2 °C/75 ± 5% RH

**Table 2 gels-08-00144-t002:** Precision and accuracy of paracetamol jelly at intra- and inter-day.

	Intra-Day			Inter-Day		
Level	Peak Area	% Recovery	% RSD	Peak Area	% Recovery	%RSD
100% spiked sample	278,093	99.01 ± 1.38	1.4	269,518	98.5 ± 1.18	1.19
	279,699			279,012		
	271,627			273,302		
	271,175			271,424		
	271,656			271,871		
	271,630			271,740		

**Table 3 gels-08-00144-t003:** The moisture content of paracetamol instant jelly in dry form stored in packaging 1 and packaging 2 at different time points (0, 2 weeks, 1, 2, and 3 months). * symbol indicates significant differences (*p*-value < 0.05).

Packaging	Storage Conditions	0 Time	2 Week	1 Month	2 Month	3 Months
Packaging 1	Real-time	2.57 ± 0.29	3.57 ± 0.29 *	6.71 ± 0.20 *	7.38 ± 0.1 *	7.48 ± 0.18 *
	Accelerated		8.71 ± 0.19 *	10.35 ± 0.16 *	13.4 ± 0.18 *	15.16 ± 0.29 *
Packaging 2	Real-time		2.6 ± 0.3	2.82 ± 0.13	2.89 ± 0.18	3.05 ± 0.12
	Accelerated		2.68 ± 0.47	2.91 ± 0.57	2.92 ± 0.23	2.89 ± 0.99

**Table 4 gels-08-00144-t004:** The paracetamol content for samples stored in packaging 1 and packaging 2 at different time points (0, 2 weeks, 1, 2, and 3 months).

Packaging Material	Sample	0 Time	2 Week	1 Month	2 Month	3 Months
Packaging 1	Real-time	103.27 ± 0.08	102.48 ± 0.60	102.37 ± 0.33	101.87 ± 0.11	101.65 ± 0.75
	Accelerated		102.50 ± 0.30	99.8 ± 1.02	93.7 ± 0.55	89.44 ± 2.22
Packaging 2	Real-time		102.37 ± 1.6	102.22 ± 1.03	101.87 ± 0.11	101.98 ± 0.04
	Accelerated		102.78 ± 2.17	102.35 ± 2.04	101.87 ± 0.12	102.08 ± 0.09

**Table 5 gels-08-00144-t005:** The shelf life of the samples stored in packaging 1 and packaging 2.

Packaging Material	Temperature °C	Slop	K_deg_ (Days ^−1^)	t_90%_	
				Days	Years
Packaging 1	30	0.00007	0.00016	651	1.78
	40	0.0007	0.0016	65	0.18
Packaging 2	30	0.00005	0.000115	913	2.5
	40	0.00006	0.000138	761	2.08

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
