# Peer review of "Stability of Paracetamol Instant Jelly for Reconstitution: Impact of Packaging, Temperature and Humidity"

_gels, 2022, doi:10.3390/gels8030144_

Round 1

Reviewer 1 Report

The article presents its relevance, however, I believe that the questions below need to be answered or added.

  1. Why were the accelerated stability tests not performed within 6 months? Would it be possible to provide?
  2. If possible, add more information about the different types of stability tests. Its main characteristics, such as time applied, controlled temperature and humidity conditions and, if possible, add the table referring to the regions and their protocols (referring to temperature).
  3. The authors need to make clear the difference between the packages present in the market and those chosen by the authors (primary or secondary). Why exactly were these chosen? What is your purpose within the study? Does paracetamol have photosensitivity? Do the packaging used protect the drug from photodegradation? How does this relate when the drug is incorporated into another delivery system or accompanied with a specific excipient, does something change?
  4. If possible, observe light, thermal and other stress conditions for a longer period than shown (Figure 2).
  5. I suggest adding tests referring to microbiological stability and correlating with the impact that the 3 variables had on these aspects.
  6. The authors did not perform the in vivo pharmacokinetic studies. These studies are relevant for increasing the impact of work.
  7. The added data are revealing, however the authors do not discuss with the literature data and do not make a correlation with the production process in the pharmaceutical industry. What are the main advantages of this drug in this pharmaceutical form compared to those produced in the industry and available to patients? Is its applicability viable? The cost of production?what are the main advantages in terms of bioavailability and dosage for patients?

Author Response

Reply to reviewers’ comments

Reviewer 1

First of all, thanks for your great comments which helped us to improve our article. Below are the responses to the reviewer comments:

Point 1:

Why were the accelerated stability tests not performed within 6 months? Would it be possible to provide?

Response 1:

Stability study was carried out in the formulation stage for lab scale batches. It provides an early indication of the product stability and thereby shortens the development plan. The prior stability testing is crucial to establish the suggested storage conditions as well as proper packaging of the product. Obviously, a complete protocol for evaluation of stability of drug substance and drug product in proposed storage conditions needs to be provided for marketing. The complete 6 months stability is preferred to be performed on a batch prepared at larger scale, representing actual industrial steps. In this study, all our experiments were on lab scale only and we are sorry that we did not proceed to 6-months’ time point.

Point 2:

If possible, add more information about the different types of stability tests. Its main characteristics, such as time applied, controlled temperature and humidity conditions and, if possible, add the table referring to the regions and their protocols (referring to temperature)

Response 2:

As suggested by the reviewer, the following information was added to the manuscript on page (2) line (66-91).

Point 3:

The authors need to make clear the difference between the packages present in the market and those chosen by the authors (primary or secondary). Why exactly were these chosen? What is your purpose within the study? Does paracetamol have photosensitivity? Do the packaging used protect the drug from photodegradation? How does this relate when the drug is incorporated into another delivery system or accompanied with a specific excipient, does something change?

Response 3:

The following text was added to the manuscript page (3) line (93-120) and the answer for select these packages in our study illustrated in Page (12), Line (373-381)

Point 4:

If possible, observe light, thermal and other stress conditions for a longer period than shown (Figure 2)

Response 4:

None of the regulatory guidelines provide exact procedure for forced degradation. Therefore, sponsor has to design stress testing protocol based on practical scientific approach. In practice, forced degradation should be started in a moderate condition and need to observe the percentage degradation [1].  As observed in Figure 2, the degradation occurred extensively in thermal stress in acid, base and oxidative condition. More time period will lead to disappear of paracetamol peak. For light, the result was in agreement with most of researcher on effect of light to paracetamol.

Point 5:

I suggest adding tests referring to microbiological stability and correlating with the impact that the 3 variables had on these aspects.

Response 5:

Most pharmaceutical companies test oral solid dosage forms for microbial bioburden on a routine basis. However, routine testing of many products is not necessary because the low water activity of these products will not promote the proliferation of microorganisms. Furthermore, various manufacturing processes for oral solid dosage forms create hostile environments for microorganisms. Pharmaceutical manufacturing processes are designed to prevent objectionable microorganisms in drug products not required to be sterile. Hence, it is expected that oral solid dosage forms will have a natural microbial load of nonobjectionable microorganisms [2]. Moreover, the stability conditions described in ICH Topic Q1A (R2) do not allow the organisms of interest in pharmaceutical solids to grow, due to either an unfavourable temperature or humidity. For this reason, testing either microbial limits or measuring water activity as part of the stability program is considered of little value in determining microbial stability [3]. Our formulation is considered among solid dosage forms because it is like a single dose dry powder to be reconstituted with water directly before use. Therefore, the present of moisture that trigger microbial grow is low comparing with liquid dosage form. The use of plastic sachets resulted in increased water content which might promote microbial growth and hence the microbial testing would have important value in determining the effect of packaging. However, owing to the fact that the preparation was in lab scale and not in a GMP environment, the microbial aspect of the batch was not critically monitored during the preparation. This makes microbial testing during storage to be of limited value and therefore, it was not performed.

Point 6:

The authors did not perform the in vivo pharmacokinetic studies. These studies are relevant for increasing the impact of work.

Response 6:

The main aim of preparing our formulation is to mask the bitter taste of paracetamol and to provide dosage form for pediatrics. Unfortunately, we did not include in vivo pharmacokinetic studies. First of all, it was not the main objective. Secondly, major regulatory authorities have introduced biowaivers for some selected medicines belonging to BCS class 1 such as Paracetamol (highly soluble and highly permeable). Comparative dissolution tests are used in biowaiver procedure to waiver the bioequivalence requirement. There are several in vivo and in vitro approaches to determine the equivalence of two pharmaceutically equivalent drug products. Major regulatory authorities such as United States Food and Drug Administration (US-FDA) and World Health Organization (WHO) have introduced biowaivers and because in vivo bioequivalent (BE) studies are time consuming and expensive to conduct, the waives in vivo BE studies by the means of comparative in-vitro dissolution test [4]. In pervious study of “Formulation development of paracetamol instant jelly for paediatric use”, The dissolution paracetamol jelly was similar to paracetamol chewable tablet (Panadol®), as similarity factor (f2) was 57.14 (greater than 50), which provided satisfactory results. Additionally, the %DE of the references and the test products can be said to be equivalent if the difference between their dissolution efficiency is within the appropriate limits (±10%, which is often used) (54). The dissolution efficiency of paracetamol suspension (Panadol®) was 93.68%, paracetamol chewable tablet (Panadol®) was 87.53% and paracetamol instant jelly was 84.15%. The dissolution efficiencies (% DE) of paracetamol instant jelly compared to the marketed products was less than 10 and can be considered as interchangeable.

Point 7:

The added data are revealing, however the authors do not discuss with the literature data and do not make a correlation with the production process in the pharmaceutical industry. What are the main advantages of this drug in this pharmaceutical form compared to those produced in the industry and available to patients? Is its applicability viable? The cost of production? What are the main advantages in terms of bioavailability and dosage for patients?

Response 7:

As suggested by the reviewer, the following information was added to the manuscript on page (1 and 2) line (33-65).

References

[1]       P. Sengupta, B. Chatterjee, and R. K. J. I. j. o. p. Tekade, "Current regulatory requirements and practical approaches for stability analysis of pharmaceutical products: A comprehensive review," vol. 543, no. 1-2, pp. 328-344, 2018.

[2]       J. E. J. P. t. Martínez, "Microbial bioburden on oral solid dosage forms," vol. 26, no. 2, pp. 58-71, 2002.

[3]       L. Skovronsky, "Inhibition of microbial growth in solid dosages at ICH stability storage conditions," ed: Recuperado, 2011.

[4]       A. D. Rathnayake, U. Mannapperuma, D. Thambawita, K. P. Herath, P. Galappatthy, and R. L. Jayakody, "Determination of in-vitro Equivalence of Paracetamol Tablets," 2016.

Reviewer 2 Report

I do not have major issues with this paper except that it does not tell anything about the formulation itself (what are these beads? previous work should be at least introduced - ref 21 I think is their work) and how it is reconstituted (water? volume? agitation? for the intended use but also in experiments presented  eg 4.2.5 TA experiment -  4.2.6 rheology)

Maybe the packaging could be shown (picture).

were  placebo or pure drug used in the stability study? if not why?

why was the drug release done at pH5.8 - what about more acidic gastric conditions?

is there enough stats to compare results eg moisture content at different time points? show significance figure 5

figure missing (c and d) in Figure 6

conclusion is very short - maybe give a bit more details of findings

Author Response

Reply to reviewers’ comments

Reviewer 2

First of all, thanks for your great comments which helped us to improve our article. Below are the responses to the reviewer comments:

Reviewer 2

Point 1:

I do not have major issues with this paper except that it does not tell anything about the formulation itself (what are these beads? previous work should be at least introduced - ref 21 I think is their work) and how it is reconstituted (water? volume? agitation? for the intended use but also in experiments presented  eg 4.2.5 TA experiment -  4.2.6 rheology)

Response 1:

More information about beads and previous work have been introduced on page (1 and 2) line (33-65) while reconstitution method has been described on page (12) line (369- 371)

Point 2:

Maybe the packaging could be shown (picture).

Response 2:

The packaging pictures have been added to manuscript: (Figure 10, Page 12, Line 386)

Point 3:

were  placebo or pure drug used in the stability study? if not why?

Response 3:

The placebo or pure drug was not included in the stability study because the purpose of this stability study is to see how the drug, in the presence of excipients, remains stable during the shelf of pharmaceutical product. A placebo may be of big vale when investigating the interaction between the excipients during the storage, but that was not our objective of this study.

Point 4:

why was the drug release done at pH5.8 - what about more acidic gastric conditions?

Response 4:

Because the phosphate buffer pH 5.8 is the dissolution media was used in pharmacopeia for paracetamol tablet. This media has been used for solubility and dissolution assessment of paracetamol are referenced in the majority of USP monographs and can be valuable and provide simple, reasonably accurate assessments of in vivo solubility and dissolution rate.

Point 5:

is there enough stats to compare results eg moisture content at different time points? show significance figure 5

Response 5:

Statistical Analysis Minitab 17 software was used to compare between the results. The texture profile analysis of most jellies in Figure 5 shows similar results to 0 time point except for jellies stored in packaging 1 at the accelerated conditions. For more clarification, the significant different samples (p-value < 0.05) will mark with * in (Figure 5, Page 8, Line 245) as well as the * mark has been added to significant different moisture content (p-value < 0.05) comparing with 0 time point in (Table 3, Page 7, Line 215)

Point 6:

figure missing (c and d) in Figure 6

Response 6:

The figure missing (c and d) has been added. (Figure 6, Page 9, Line 270)

Point 7:

conclusion is very short - maybe give a bit more details of findings

Response 7: 

Conclusion has been rewritten in (Page 11 and 12), and Line (383-352).

Round 2

Reviewer 1 Report

Thank you for the clarifications, therefore, I consider the manuscript suitable for publication.